# Dielectrophoretic and Electrical Impedance Differentiation of Cancerous Cells Based on Biophysical Phenotype

**DOI:** 10.3390/bios11100401

**Published:** 2021-10-17

**Authors:** Ina Turcan, Iuliana Caras, Thomas Gabriel Schreiner, Catalin Tucureanu, Aurora Salageanu, Valentin Vasile, Marioara Avram, Bianca Tincu, Marius Andrei Olariu

**Affiliations:** 1Department of Electrical Measurements and Materials, Faculty of Electrical Engineering and Information Technology, Gheorghe Asachi Technical University of Iasi, 21-23 Profesor Dimitrie Mangeron Blvd., 700050 Iasi, Romania; ina.turcan@student.tuiasi.ro (I.T.); thomas.schreiner@umfiasi.ro (T.G.S.); 2“Cantacuzino” National Medical-Military Institute for Research and Development, 103 Splaiul Independentei, 050096 Bucharest, Romania; caras.iuliana@cantacuzino.ro (I.C.); tucureanu.catalin@cantacuzino.ro (C.T.); salageanu.aurora@cantacuzino.ro (A.S.); vasile.valentin@cantacuzino.ro (V.V.); 3Faculty of Medicine, “Grigore T. Popa” University of Medicine and Pharmacy, 16 Universitatii Street, 700115 Iasi, Romania; 4National Institute for Research and Development in Microtechnologies—IMT Bucharest, 126A Erou Iancu Nicolae Street, 077190 Bucharest, Romania; marioara.avram@imt.ro (M.A.); bianca.tincu@imt.ro (B.T.); 5DDS Diagnostic SRL, 7 Vulcan Judetu Street, 030423 Bucharest, Romania; 6Faculty of Applied Chemistry and Material Science, University “Politehnica” of Bucharest, 313 Splaiul Indepentei, 060042 Bucharest, Romania

**Keywords:** cancer cells, dielectrophoresis, crossover frequency, electrical impedance spectroscopy

## Abstract

Here, we reported a study on the detection and electrical characterization of both cancer cell line and primary tumor cells. Dielectrophoresis (DEP) and electrical impedance spectroscopy (EIS) were jointly employed to enable the rapid and label-free differentiation of various cancer cells from normal ones. The primary tumor cells that were collected from two colorectal cancer patients, cancer cell lines (SW-403, Jurkat, and THP-1), and healthy peripheral blood mononuclear cells (PBMCs) were trapped first at the level of interdigitated microelectrodes with the help of dielectrophoresis. Correlation of the cells dielectric characteristics that was obtained via electrical impedance spectroscopy (EIS) allowed evident differentiation of the various types of cell. The differentiations were assigned to a “dielectric phenotype” based on their crossover frequencies. Finally, Randles equivalent circuit model was employed for highlighting the differences with regard to a series group of charge transport resistance and constant phase element for cancerous and normal cells.

## 1. Introduction

Label-free manipulation, sorting, and isolation of biological cells and, in particular, cancerous cells, are a matter of concern at a worldwide level. Dielectrophoresis (DEP) has emerged as a potential technique for this purpose since early 90’s. To date, DEP has been extensively employed in the electromanipulation of cancer cells and many studies can be provided as examples of best practices. The increased interest in DEP-based technique utilization is justified mainly by the fact that the technique does not require prior knowledge of specific cells, as it does in biomarker related isolation techniques. Capturing cells on DEP systems has the enormous advantage of reversibility, maintaining the cells viability for further characterization and culturing. Initial experimental DEP-based studies focused on the manipulation, sorting, and isolation of cancer cells. DEP experiments that focused on various types of cancer cells have been already reported: breast, MCF-10A, MCF-7, MDA-MB-231 and MDA-MB-435 [1,2,3]; oral, HOK, H357, H157 [4,5]; leukemia, K526 [6]; kidney, HEK 293, 786-O [7,8]; ovary, SKOV-3 [3,9]; prostate PC3, LnCap [3]; lung A549, H1299, 95C, 95D [10,11]; cervical, HeLa [12]; and colorectal, HCT-116 [2]. The aforementioned studies highlight the possibility of discriminating cancerous cells on the basis of their crossover frequencies. Recently, Turcan and Olariu [13] presented in a centralized manner, the evolution of dielectric parameters versus crossover frequencies.

On the other hand, biophysical characterization of cancer cells on the basis of impedance measurements has been studied with the aim of identifying the “electrical signature” of various types of cancers, which may allow for the label-free successful evaluation of therapeutic efficiency. Both the impedance flow cytometry (IFC) and the electrical impedance spectroscopy (EIS) were used as techniques to gather impedance data for bulk cell suspensions, clustered cells, or single cells. EIS is characterizing the evolution of dielectric parameters against the frequency as a result of the interaction between an electrical stimulus (i.e., external electric field) and the biological matter. The dielectric behaviour of polarized cells is analysed with respect to the evolution of three (α-, β-, and γ-) dispersions of different magnitudes which may occur at different frequencies.

Human cancers consist of cells that display different phenotypic features, including cellular morphology, gene expression, metabolism, motility, proliferation, and metastatic potential [14]. This heterogeneity is a result of the interplay between cell-intrinsic (i.e., the variability in the genetics, epigenetics, and the biology of a tumor’s cell-of-origin) and cell-extrinsic factors (i.e., those arising from factors in the microenvironment) which shape the cellular phenotype [15,16]. Consequently, the phenotypic heterogeneity within tumors constitutes a major impediment in their diagnostics and therapy. From this point of view, EIS has the ability to monitor the dynamics of intrinsic and extrinsic changes that occur in cancer cells [17].

Breast mammalian cancer cells are among the most studied cancer cells in dielectric studies [18,19,20,21]. Qiao et al. [18] employed electrical impedance spectroscopy for monitoring a cells’ state in solutions. The measurements were performed between 300 kHz and 1 MHz at the level of MDA-MB-435S, MDA-MB-231, MDA-MB-7, and MCF-10A cell lines. The characteristic relaxations increased from normal to late cancer stages which allowed clear differentiation of each cell’s electrical signature. Moreover, Huerta-Nuñez reported the [19] successful identification of breast cancer with the help of impedance spectroscopy by performing studies on solutions of non-metastatic (MCF-7, MDA-MB-231) and metastatic (SK-BR-3) breast cancer cells that were coupled with magnetic nanoparticles of very low concentrations.

The comparative impedance measurements on lung and liver cancer cells were reported by Al Ahmad [22] who highlighted the reduced ability of cancerous cells for storing energy in comparison to normal cells. In [23], Zhang reported not only the capability of impedimetric measurements for distinguishing between skin cancer cells (A431) and normal cells (HaCaT), but also confirmed the capacity of the technique of providing real-time kinetic information on cell proliferation behaviour.

Therefore, electromanipulation and electrical characterization of cancerous cells has demonstrated good differentiation among various types of cells from an electrical viewpoint. A much more powerful tool may be developed by combining EIS with DEP. Nguyen et al. concentrated A549 cells while applying p-DEP (positive DEP) at the level of circular microelectrodes at the frequency of 1 MHz and the potential of 10 Vpp (peak-to-peak voltage) [10]. The impedance measurements that were performed demonstrated a linear relation between the impedance variation and the cells’ number, and therefore, the high potential of the technique for being employed when there is a low quantity of cells. Thus, combined exploitation of EIS and DEP provides Appendix A on cancer cell dielectric properties and correlations to their biophysical phenotype can be made.

In this paper, we are reporting on the utilization of DEP for trapping cancer cell lines as well as primary tumor cells. All the cells were firstly suspended in a low conductivity suspension medium and concentrated with the help of dielectrophoresis at the level of interdigitated (castellated) microelectrodes. The differentiation among the different types of cancer cells (including primary tumor cells that were collected from two colorectal cancer patients and cancer cell lines (SW-403, Jurkat and THP-1)), and healthy peripheral blood mononuclear cells (PBMCs) was done based on EIS experiments following DEP cells’ trapping and identification of crossover frequencies for each type of cell.

## 2. Materials and Methods

### 2.1. Fabrication of Interdigitated Microelectrodes

The interdigitated microelectrodes were manufactured within a clean room facility class 1000 (ISO 5). Metal-based microelectrodes were fabricated by a lift-off technique on an oxidized 4-inch silicon (Si) wafer, using photoresist as a sacrificial layer. LOR 10B photoresist was spin coated on top of the Si wafer at 3000 rpm for 30 s and pre-baked on a hot plate at 150 °C for 5 min, followed by a spin coating of HPR 504 photoresist at 2000 rpm for 30 s and pre-baking on a hot plate at 95 °C for 45 s. The two photoresists were imprinted by UV lithography; exposure was performed in a MA6 mask aligner (Suss MicroTec) for 2.5 s to transfer the pattern from the photolithographic mask to the photoresist. Following UV exposure, the photoresist was developed in a specific solution (HPRD 402) for 30 s. In this step, the two photoresists were patterned with the layout of the conductive electrodes where the UV exposure was performed.

Metal deposition was performed in an e-beam evaporator (Neva 005). First, a 50 nm layer of titanium was used to promote adhesion, then a 500 nm thin gold film was deposited. The lift-off process was completed in acetone to allow the photoresist to dissolve while leaving behind the metal pattern. This process was used for electrode gold patterning on the surface of Si wafer. Wafer cleaning was performed in a solvent mixture (acetone, isopropyl, and deionized water) at boiling temperature.

To obtain a passivation of the gold conductive lines, the SU-8 2015 was spin coated at 3000 rpm for 30 s and then pre-baked at 65 °C for 1 min and 95 °C for 5 min. The SU-8 resist was exposed using a photolithographic mask using the MA6 mask aligner for 8 s, followed by post-baking on a hot plate at 65 °C for 1 min and then at 95 °C for 6 min, and developed in mr-DEV-600 solution for 2 min. The wafer was then washed in isopropyl alcohol to stop the action of the developer. To guarantee that the SU-8 passivation layer properties did not modify, the wafer was hard baked on a hot plate at 180 °C for 10 min. The designed microelectrodes on the Si wafer were drawn and cut individually.

The geometry of the interdigitated microelectrodes was tailored in accordance with the cells under study. A castellated architecture was selected for ensuring the development of higher gradient field regions. Each interdigitated microelectrode array had 16 fingers with a length of 2560 μm, the gap between the fingers and the intercastellations had a dimension of 40 μm.

### 2.2. Cell Culture and Sample Preparation

#### 2.2.1. Cell Lines

Human colon adenocarcinoma cell line SW-403 (Cat. No. 87071008), human leukemic T cell line Jurkat E6.1 (Cat No. ECACC 88042803), and human monocyte-like THP-1 cells (Cat. No. 880881201) were purchased from the European Collection of Authenticated Cell Cultures (ECACC) and cultured in RPMI-1640 (Bio Whittaker Lonza, Verviers, Belgium), supplemented with 10% fetal bovine serum (FBS, Euroclone, Milan, Italy) and 100 IU/mL penicillin + 100 µg/mL streptomycin (Lonza, Basel, Switzerland) (complete culture medium). Cell lines were incubated at 37 °C in an atmosphere supplemented with 5% CO_2_, in 75 cm^2^ flasks. The adherent cell line, SW-403, was cultured until 85% confluence, then washed with phosphate buffered saline (PBS, Merck, Darmstadt, Germany), and detached using 0.05% trypsin-EDTA solution (Thermo Fischer Scientific, Waltham, MA, USA). The cells were then suspended in a complete culture medium, washed by centrifugation at 200× g for 10 min and then resuspended in a fresh complete culture medium. Non-adherent cell lines (THP-1 and Jurkat) were simply collected and centrifuged in the previously mentioned conditions.

#### 2.2.2. Isolation and Culture of Primary Tumor Cells

Tumor samples (T1 and T2) were collected from two colorectal cancer patients after written informed consent from each subject and approval from the Ethics Committees of Bucharest Emergency University Hospital and processed as previously described [24]. Briefly, the tumor specimens were excised carefully and aseptically during surgery and transferred to 50 mL tubes with PBS supplemented with antibiotics (100 IU/mL penicillin, 100 µg/mL streptomycin, 1 mg/mL gentamicin, and 0.5 mg/mL vancomycin). Tumor tissues were then transferred to Petri dishes and rinsed with fresh AIM-V containing AlbuMAX^®^ supplement (bovine serum albumin) medium (Thermo Fischer Scientific, Waltham, MA, USA). After resection of the fatty and connective tissues and the necrotic areas, the tumor specimens were minced with sterile scalpels and scissors into small pieces (0.5–1 mm^3^) and cultured in AIM-V AlbuMAX supplemented with antibiotics (100 IU/mL penicillin, 100 µg/mL streptomycin, 20 µg/mL gentamicin (Merck, Darmstadt, Germany), and 6 µg/mL vancomycin (Merck, Darmstadt, Germany)) and amphotericin B (5 µg/mL) (Merck, Darmstadt, Germany). The primary tumor cells were maintained in culture continuously for more than 12 months. The cancer cells grew as floating spheroids/aggregates, firmly/loosely adherent spheroids, or as both adherent and floating spheroids/aggregates. Subsequent passages were performed every two or four weeks. To obtain single cells, spheroids/aggregates were dissociated by enzymatic digestion using Accumax-Cell aggregate dissociation medium (Thermo Fischer Scientific, Waltham, MA, USA).

#### 2.2.3. Human Peripheral Blood Mononuclear Cells

Human blood was obtained from a healthy donor (lab worker) after obtaining informed consent and ethical approval. Peripheral blood mononuclear cells (PBMCs) were isolated by using Ficoll-Hypaque (1.077 g/mL density, (Merck, Darmstadt, Germany)) and resuspended in RPMI medium.

#### 2.2.4. Suspension Medium

The low conductivity suspension medium (250 mM sucrose, 13 mS/m conductivity) was chosen based on viability data in preliminary experiments and was prepared by dissolving sucrose (Merck, Darmstadt, Germany) in distilled water and adjusting the pH to 7.4. The osmolarity was measured with a VAPRO Vapor Pressure Osmometer Model 5600 and was 250 mmol/kg. The conductivity measurement was performed with a ZetaSizer Nano-2S. The baseline value was 0.5 mS/m. To increase the conductivity to 13 mS/m, a 250 mM HEPES (Merck, Darmstadt, Germany) solution was used. The pH and conductivity values remained stable for at least one week when stored at 4 °C.

#### 2.2.5. Sample Preparation and Viability Assay

Cancer cell lines, primary tumor cells, and normal PBMC were washed and resuspended in a low conductivity suspension medium (2 × 10^6^ cells/mL). Their viability was evaluated before and after DEP measurements by staining the cells with acridine orange (Merck, Darmstadt, Germany) and propidium iodide (Merck, Darmstadt, Germany) and examining them in a fluorescence microscope (Nikon TE2000) at 100X magnification. Cells that were fluorescing green were scored as viable while cells that were fluorescing orange, either fully or partially, were scored as nonviable.

### 2.3. Experimental Set-Up and Equipment

The experimental activity in this study involved trapping the cells via dielectrophoresis to determine the crossover frequency by observing the cells’ motion and characterization of the trapped cells via electrical impedance spectroscopy. Figure 1 depicts a schematic diagram of the proposed experimental structure. The set-up operation procedure involved two main steps: (1) trapping the cells at the microelectrode level via DEP and (2) identification of the cell’s type by measuring its impedance characteristics. The test section of the microchip consisted of the electrode substrate on the bottom and a glass cover on the top for observation of the cells. For the DEP experiments, a Keysight 33521A Function/Arbitrary Waveform Generator was employed to generate a sinusoidal AC electric field. The cell’s distribution at microelectrode level was monitored and recorded using an improvised optical setup consisting of a Nikon Plan Fluor 10x/0.30 microscope objective with a mounted CCD Nikon Digital Sight DS-Qi1Mc camera connected to a computer that was running NIS-Elements AR 3.0 SP 1 (Build 455) software. Electrical impedance spectroscopy measurements were performed using a Novocontrol Broadband Dielectric Spectrometer (Alpha-A High-Performance Frequency Analyzer). The electrodes were connected to the analyser and generator by using a Micrux drop-cell connector. The impedance experimental data were fitted with the software EIS Spectrum Analyser 1.0 program [25].

## 3. Results and Discussion

### 3.1. DEP-Based Cells Manipulation and Electrical Impedance Spectroscopy Measurement

In the presence of an inhomogeneous electric field gradient, the biological cells may be displaced towards the electric field maxima (positive DEP) or towards the electric field minima (negative DEP) depending on the dielectric properties of the specific cell and on the properties of the suspending media. In a first experiment, normal (PBMC) and tumor cells (SW-403 cell line and primary tumor cells T1) were subjected to a sinusoidal excitation voltage (9 V peak-to-peak magnitude and a frequency of 1 MHz) that was applied to the electrodes, for approximately 5 min to concentrate the cells on the electrodes. Under the effect of p-DEP, after few seconds (≈4 s) the cells concentrated at the electrode surface (see the Appendix A). Figure 2 depicts the microscopic images of cell samples before and after (5 min) DEP manipulation. It is visible that under these experimental conditions the majority of the cells of both normal and tumor cell populations were displaced towards the highest electric field regions. Moreover, in some regions the cells followed electric field lines between adjacent microelectrodes due to their high interfacial polarization, creating “cells’ bridges”. Before and after the cells were trapped at the level of microelectrodes, the impedance measurements were carried out to differentiate the normal cells from cancer ones.

Next, the impedance measurements were performed on normal cells (PBMC and THP-1-monocyte cell line), two tumor cell lines (an adherent adenocarcinoma cell line (SW-403) and Jurkat, a non-adherent T cell line), and the primary tumor cells isolated from two colon cancer patients (T1 and T2).

The impedance measurements of the un-trapped and trapped living cells were performed in the frequency range from 0.1 to 300 kHz at an operating voltage of 100 mV. This frequency range was selected to monitor the evolution of the electrical properties of each cell type in the α and β dispersion regions. The frequency range was selected for exploring the effect of ionic diffusion and interfacial polarization of biological membrane systems [26]. Figure 3 depicts the measured electrical impedance spectra (amplitude Z, phase angle θ, and Nyquist plots) of the three cell types (cancer cell lines, primary tumor cells, and normal PBMCs) before and after the DEP concentration at the electrode level. The impedance magnitude of the suspension medium (in the absence of cells) decreased when the frequency increased. The transition from capacitive behaviour, which dominates at lower frequencies, to the resistive behaviour, that prevails at higher frequencies, was highlighted. Generally, adding the cells to the suspension medium lead to an augmentation of the total impedance as compared to the medium alone (Figure 3a).

However, before DEP manipulation, when the cells are suspended within the entire volume of the suspension solution, no significant differences among the different types of cells could be noticed. The presence of each cell sample uniquely changed the impedance response of the suspension medium. By contrast, in the presence of DEP forces that were applied for 5 min, a different feature below 10 kHz was noticed on the phase angle and Nyquist (Figure 3e,f) characteristics. The THP-1 cell line is regarded as a model for primary monocytes (i.e., monocytes from human peripheral blood), which explains their similarity in terms of dielectric responses. As a consequence of cell migration onto the electrode surface, the local ionic environment at the electrode/electrolyte interface was affected due to high insulating of the cell membranes; cell trapping lead to a decrease of electrode surface area and therefore an increase of the interface impedance. At low frequencies, the current was forced to flow between the insulating cell membranes, while at higher frequencies the current penetrated the cell membranes and flowed through the intracellular and extracellular fluid [27,28]. Therefore, differences noticed at frequencies below 10 kHz between normal cells (PBMC and THP-1) and cancer cells (Jurkat, SW-403, T1, and T2) may be attributed to the surface morphological features of the cell membranes and to the electrode surface area which is covered with cells (i.e., the cells radii). At higher frequencies, the spectrum of the total impedance is presumably influenced by the suspension medium, reaching almost the same value for all cell types, as can be seen in Figure 3d.

The functionality and reproducibility of the proposed method was evaluated from the EIS responses of trapped cells at different DEP operating voltages with five independent interdigitated microelectrodes that were fabricated by a similar procedure. Based on the impedance magnitude and the phase angle frequency dependences that are illustrated in Appendix A, no significant differences in the EIS responses were observed in all five individual microelectrodes that were employed for trapping T2 cancer cells at 9 Vpp and 1 MHz. Appendix A depicts the average values of the impedance magnitude and phase angle, the standard deviation (SD), and the relative standard deviation (RSD) that was calculated between the electrodes at three different frequencies (10^3^, 10^4^, and 10^5^ Hz). The reproducibility of our manufactured interdigitated microelectrodes was found to be good with a%RSD yield in the range of 2.24 to 6.44%. Furthermore, method reproducibility was evaluated by analyzing the influence of the DEP operating voltages (3, 6, 9, and 12 Vpp) on the electrical characterization of PBMC, THP-1, and T2 cells (Appendix A). Even if the DEP voltage was changed, the impedance spectra were similar with minor variations at low frequencies in the case of PBMC and T2 cells obtained from donors due to their heterogeneity.

To understand which specific characteristics influenced the different features of the normal and cancer cells, the DEP crossover frequency experiment and the electrical equivalent circuit model were used.

The DEP crossover frequency ($f_{co}$) is the characteristic frequency at which the polarity of the dielectrophoretic force changes and cells experience zero DEP force. By observing the motion of cells at the electrode edges when the frequency that is applied is slowly swept, the $f_{co}$ of each cell type can be ascertained [3,29]. In our dielectrophoretic crossover frequency experiment, the microchip was powered by an AC voltage with 12 Vpp of variable frequency at the level of two adjacent microelectrodes. It should be mentioned that the DEP operating voltage was not affecting the impedance spectra (please see the Appendix A) when the experiments were running during the same period of time, however, for the crossover frequency experiments we choose 12 Vpp voltage as the displacement of the cells is more visible. The voltage frequency was sequentially increased from 10 kHz up to 1 MHz and the cell displacements induced by the DEP force were examined with a microscope. The crossover frequency at which the cell exercised no DEP movement was recorded. Within individual experiments, at least 10 frequencies were determined for each cell type and all measurements were performed at room temperature.

Figure 4 depicts the experimentally determined crossover frequencies for various human cancer cells, including the primary tumor cells (T1 and T2) that were collected from two colorectal cancer patients, a colon adenocarcinoma cell line (SW-403), a human leukemic T cell line (Jurkat), a human monocyte-like cell line (THP-1), and peripheral blood mononuclear cells (PBMCs) from a healthy subject, that were all suspended in medium with a conductivity of 13 mS/m. As expected, THP-1, Jurkat, and SW-403 cancer cell lines exhibited distinct behaviour, characterized by lower average crossover frequencies (57.4 ± 2.5 kHz, 31.6 ± 1.7 kHz, and 28.2 ± 1.4 kHz, respectively) in comparison to PBMCs (106.2 ± 5.4 kHz), which allowed discrimination of each type of cell. Moreover, the primary tumor cells (T1 and T2) presented characteristic crossover frequencies within the same domain of frequency as also observed for the cancer cell lines. According to the literature, these different DEP frequency responses of cancer and normal blood cells may be explained and expressed by Gascoyne and Shim [7] in terms of reciprocal cell “dielectric phenotype” 1/Rϕ, where ϕ represents the membrane folding factor (the ratio of actual membrane area to that of the idealized smooth shell) and R is the cell radius. Many studies have reported that cancer cells have a larger folding factor and radii than both blood cells and normal cells of comparable origin [4,5,8,30,31,32,33,34]. A plausible explanation could be related to an increase in the membrane cholesterol or the membrane lipid rafts in cancer cells [35,36].

Due to the notorious heterogeneity of cancer cells, especially of the primary tumor cells, it was difficult to estimate *R* and *ϕ* parameters for each cell type. Thus, the following discussions are based on the reciprocal dielectric phenotype which is proportional to the DEP crossover frequency:(1)$$f_{co}\approx \frac{1}{\sqrt{2}\pi C_{0}}\left(\frac{\sigma_{s}}{R\phi }\right)$$
where $\sigma_{s}$ is the conductivity of the suspending medium and $C_{0}=9\;mF/m^{2}$ [37] represents the specific capacitance of the smooth cell plasma membrane. The calculated reciprocal cell dielectric phenotype (Appendix A) demonstrated notable differences between the cancer and normal peripheral blood mononuclear cells, highlighting the fact that the dielectric response of each cell type is influenced significantly by its morphological characteristics (i.e., its size and shape).

### 3.2. Interpretation of Measured Impedance Data by Equivalent Circuit

To explain the electrical impedance characteristics of the cell-covered electrode, an electrical equivalent circuit model was used. The experimental impedance spectra (Nyquist plots) were analysed in accordance to Randles equivalent circuit model [38] (Figure 3f, inset). The electrolyte’s resistance, $R_{S}$, represents the suspension medium in series with a parallel group of double layer capacitance $C_{DL}$ necessary for the charging of the electrode/electrolyte interface and charge transport element, that is represented by a series group of charge transport resistance $R_{CT}$ and a constant phase element CPE.

Since cell membranes, ideally modelled as capacitors, include a lipid bilayer, surface roughness, and integrated ion channels that resemble a porous surface contact, the capacitance was modulated by the charge transfer and differs from the capacitance of an ideal capacitor (i.e., frequency-dependent) [39,40]. Therefore, the $R_{CT}$ and CPE series group is describing the transport phenomena near the electrodes [40,41,42] (i.e., the charge transport through the electrode/electrolyte interface including the cells membrane capacitance (electrode–cells–suspension medium assembly)). Under these considerations, the total measured impedance Z of the system can be expressed as:(2)$$Z=R_{S}+\frac{Z_{DL}\left(R_{CT}+Z_{CPE}\right)}{Z_{DL}+R_{CT}+Z_{CPE}}$$
where ZDL=1jωCDL and ZCPE=1Q(jω)n where Q is a measure of the magnitude of *Z_CPE_*, ω is the angular frequency, and n is a constant (0≤n≤1).

By fitting the impedance measurements after DEP trapping, as we expected, the extracted resistance of the solutions and double layer capacitances were similar for all of the types of cells involved in our experiment, with an average value of $R_{S}=267\pm 7.8\;\Omega$ and $C_{DL}=342\pm 16.9\;\mathrm{pF}$. As shown in Figure 5a, the extracted charge transport resistances $R_{CT}$ of tumor cells T1 and T2 were approximately equal but their values were lower than the ones of normal PBMC cells. It was noticeable that the value of parameter $R_{CT}$ for PBMC cells was higher in comparison to values of the cancerous lines that were involved in the study even if, in the case of THP-1, the difference was not considerable. The extracted magnitudes Q and n constant of *Z_CPE_* for the cancer cells were in the range of $4.0\times 10^{-8}\pm 1.16\times 10^{-9}$–$6.5\times 10^{-8}\pm 2.07\times 10^{-9}$ $s^{n}/\Omega$ and 0.830–0.855, respectively, while those for the normal cells were in the range of $1.7\times 10^{-7}\pm 8.78\times 10^{-9}$–$1.9\times 10^{-7}\pm 8.35\times 10^{-9}$ $s^{n}/\Omega$ and 0.762–0.763, respectively (Figure 5b,c). Moreover, the values of Q and n of PBMC cells were very different in comparison to the values of the same parameters of Jurkat, SW-403, T1, and T2 tumor cells but close to the values for THP-1 cells. The less evident difference between the PBMC and THP-1 may be attributed to the fact that, as stated in Section 3.1, the THP-1 cell line is regarded as a model for primary monocytes. Moreover, the fact that under p-DEP, cells migrated towards the electrode interface, as is visible in Figure 2, so all charge transport phenomena at this interface is mediated and altered by these cells. Thus, the Randles circuit transport elements, $R_{CT}$ and CPE, were influenced by the cells’ size and morphological characteristics (i.e., their dielectric phenotype), especially the cell membrane features since they facilitated all of the charge transport to and from the extracellular medium.

## 4. Conclusions

We reported a study proposing the combined utilization of EIS and DEP for enabling the rapid and label-free differentiation of various cancer cells from normal ones. The method’s successful exploitation was based on the correlation of impedance characteristics of the cells with their biophysical phenotype. Experiments were performed using interdigitated microelectrodes and included three cancerous cell lines, two types of primary tumor cells, and normal blood cells. Crossover frequencies that were determined during the application of DEP forces between different types of cells achieved reasonably different values. The impedance spectra after DEP trapping demonstrated that an electrical signature may be a future solution in differentiating cancer cells from normal cells. Moreover, the Randles equivalent circuit model highlighted differences between a series group of charge transport resistance and constant phase elements for cancerous and normal cells fact which were assigned to a dielectric phenotype. Through its high capacity for discrimination, the proposed method could be a valuable approach for the detection of circulating tumor cells (CTCs).

## Figures and Tables

**Figure 1 biosensors-11-00401-f001:**
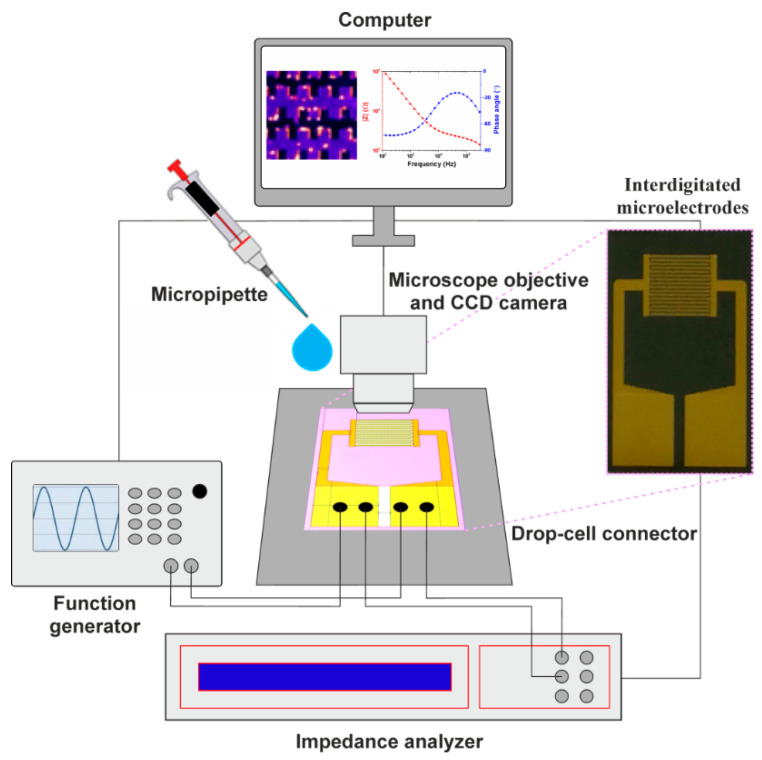
Experimental set-up.

**Figure 2 biosensors-11-00401-f002:**
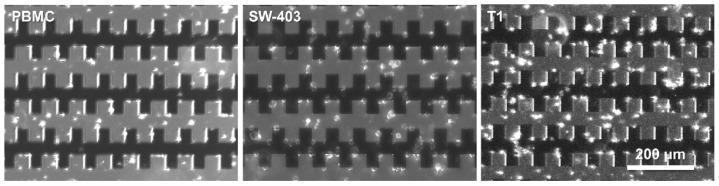
Microscopy images of a cell samples (PBMC, SW-403 and T1) distributions after DEP manipulation.

**Figure 3 biosensors-11-00401-f003:**
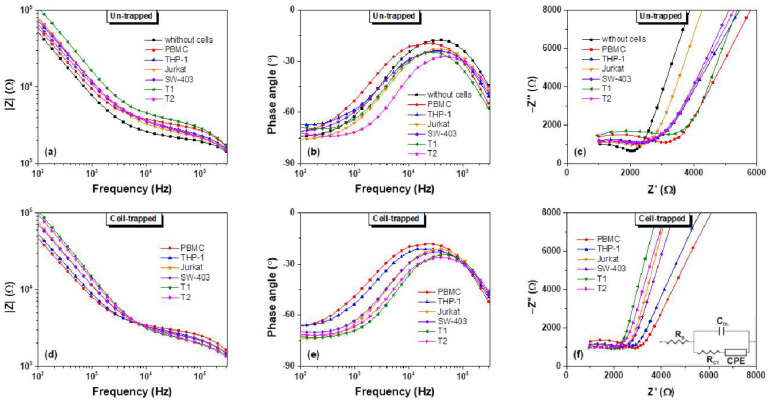
Electrical impedance spectroscopy responses (amplitude Z (**a**,**d**), phase angle (**b**,**e**), and Nyquist plots (**c**,**f**)) of different cell types suspended in buffered sucrose solution, before and after cells trapping.

**Figure 4 biosensors-11-00401-f004:**
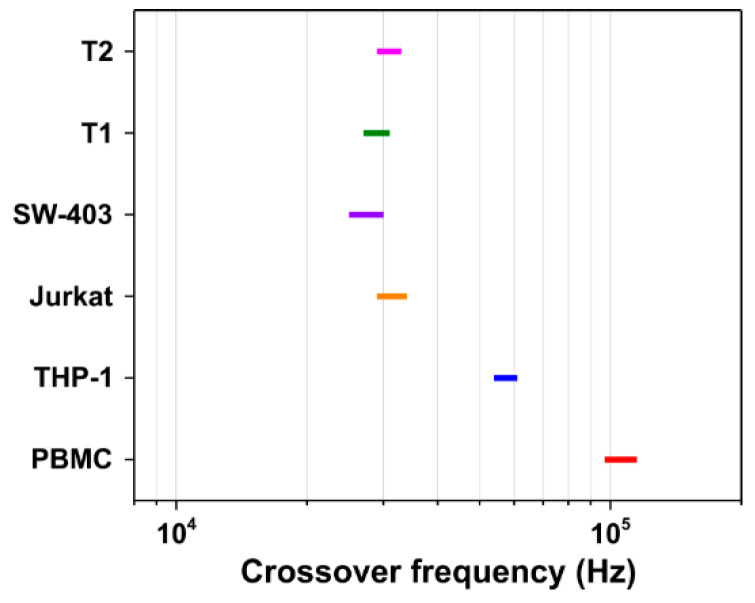
The DEP crossover frequency for the different types of cancer cells and healthy peripheral blood mononuclear cells.

**Figure 5 biosensors-11-00401-f005:**
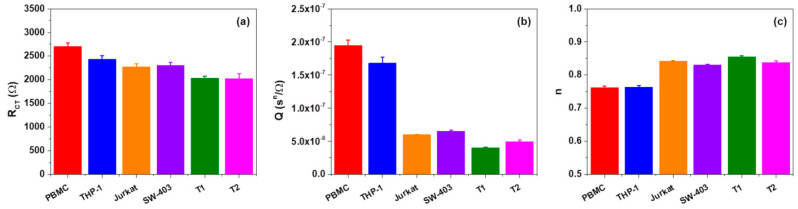
The electrical fitting parameters ($R_{CT}$ (**a**), Q (**b**), and n (**c**)) in the equivalent circuit model for studied cells. Error bars indicate the values of the relative estimated errors of the calculated parameters.

## Data Availability

The data that support the findings of this study are available on request from the corresponding author.

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
