# Peer review of "Dielectrophoretic and Electrical Impedance Differentiation of Cancerous Cells Based on Biophysical Phenotype"

_biosensors, 2021, doi:10.3390/bios11100401_

Round 1

Reviewer 1 Report

The manuscript is well written and I suggest only a minor revision of the manuscript.

The authors need to address the following questions/comments:

  1. What kind of difference shall we see in Fig. 2? I do not see any significant differences between images shown as a) or b).
  2. Redo fig. 3 to show an EIS of untrapped and cell trapped device for each particular cell line in a single figure i.e. Fig. 3a should show untrapped and trapped PBMC, while Fig. 3b should show untrapped and trapped THP-1 cells, and so on. In that way, we could see differences in spectra.  
  3. The authors need to do a principal component analysis by combination of parameters shown in Fig. 5 to provide the discrimination power of the approach for identification of cells.

Author Response

Dear Reviewer,

Thank you very much for the valuable comments and suggestions regarding the manuscript ID biosensors-1385263, entitled: “Dielectrophoretic and electrical impedance differentiation of > cancerous cells based on biophysical phenotype”.

We carefully checked the attached copy of the revised manuscript, and all the changes are clearly marked with red font.

Our point-by-point response to your comments is as follows:

  1. What kind of difference shall we see in Fig. 2? I do not see any significant differences between images shown as a) or b).

R1: Dear reviewer, you are right, the differences between Fig. 2a and 2b are not very visible, thus, we decided to remove Fig. 2a) and include in the manuscript only Fig. 2b). For additional information, within the supplementary info a video showing the difference between 0V and 9V can be easy observed.

  1. Redo fig. 3 to show an EIS of untrapped and cell trapped device for each particular cell line in a single figure i.e. Fig. 3a should show untrapped and trapped PBMC, while Fig. 3b should show untrapped and trapped THP-1 cells, and so on. In that way, we could see differences in spectra.  

R2: Please find enclosed the graph requested plotting for certain solution the trapped and un-trapped cells uploaded document). Should you consider needed we can include them within the manuscript. However, from our viewpoint, these figures are not relevant to our study as our scope was the one of differentiating various types of biological cells via DEP+EIS.

  1. The authors need to do a principal component analysis by combination of parameters shown in Fig. 5 to provide the discrimination power of the approach for identification of cells.

R3: Unfortunately, the results we have at the time being are not sufficient for performing PSA but we do consider that we achieved the aim of the report, mainly, the one of demonstrating the operability/functionality of DEP+EIS at the level of  tumor cells ( not only tumor cell lines but, more importanly, primary tumor cells) fact which in our opinion is successfully achieved.

We hope with all the above responses will be suitable for publication in MDPI Biosensors.

Sincerely yours,

Dr. Marius Andrei Olariu

Reviewer 2 Report

The paper report DEP+EIS experimental results, but: i) information is not clearly exposed, and this makes the work hard to follow; ii) the paper lacks a thorough analysis to make clear the improvement/advantages of cancerous cells detection based on the combined use of DER and EIS techniques. FoMs/metrics to parametrize this improvement also would be welcomed.

Other comments:

Technical details must be included all through the work. For instance, Section 2, 2.1 Electrodes: include detailed photograph; they are designed in a standard commercial CMOS process or in own clean room? Justify fingers length and separation. Can you assure reproducibility?  

Define clearly the cell samples used in each section: Fig. 2 use PBMC, SW-403, T1; then Fig 3 use T1, T2, THP-1, Jurkat, ….

In DEP, why a 9Vpp 1 MHz signal is used? Can lower amplitude/lower frequency be used? For the crossover frequency experiment why the amplitude is increased to 12Vpp? These values can affect the following impedance characterization?

Include photograph of the experimental setup.

Comparison with other recent works should be included to make clear the contribution of the work. The results must be not only qualitative, but FoMs/metrics to parametrize the achieved detection improvement should be introduced.

Author Response

Dear Reviewer,

Thank you very much for the valuable comments and suggestions regarding the manuscript ID biosensors-1385263, entitled: “Dielectrophoretic and electrical impedance differentiation of > cancerous cells based on biophysical phenotype”.

We carefully checked the attached copy of the revised manuscript, and all the changes are clearly marked with red font. (please see the attached document)

Our point-by-point response to your comments is as follows:

Electrodes: include detailed photograph; they are designed in a standard commercial CMOS process or in own clean room?

The electrodes were manufactured within the facilities of National Institute for Research and Development in Microtechnologies – IMT Bucharest, 126A Erou Iancu Nicolae Street, 077190, Bucharest, Romania, in their own clean room facility class 1000 (ISO 5). We specified this in the revised manuscript Section 2.1.

Justify fingers length and separation. Can you assure reproducibility?

The commercial electrodes available on the market are provided with fingers of very small width (5 or 10um). Considering the fact that the dimensions of the cells involved in the study were higher fact which will not ensure DEP operability, we decided to manufacture our own electrodes with larger width (50um) fact which is ensuring an optimal electric field gradient for allowing successful displacement of cancerous cells under DEP forces. Should you consider this info needed in the manuscript we can include it.

Define clearly the cell samples used in each section: Fig. 2 use PBMC, SW-403, T1; then Fig 3 use T1, T2, THP-1, Jurkat, ….

We include a description of the cell samples under tests in section 3.1.

In DEP, why a 9Vpp 1 MHz signal is used?

For trapping the cells at the level of electrodes lower and higher voltages than 9Vpp can be used, but the 1MHz frequency was chosen in respect to the data available in literature, collected and plotted within Fig. 13 from the paper “Turcan, I.; Olariu, M.A. Dielectrophoretic Manipulation of Cancer Cells and Their Electrical Characterization. ACS Comb. Sci. 2020, 22, 554–578, doi:10.1021/acscombsci.0c00109.” In such a manner we assured ourselves that the cells are to be submitted under positive dielectrophoretic force.

Can lower amplitude/lower frequency be used? For the crossover frequency experiment why the amplitude is increased to 12Vpp?

We choose the amplitude of 12Vpp as of the fact that we noticed that the displacement at this frequency is faster even if the displacement is observed at other amplitudes as well.

These values can affect the following impedance characterization?

The DEP magnitude is not affecting the impedance spectra (please see the figure above). For the crossover frequency experiments we choose 12Vpp voltage as the displacement of the cells is more easy visible.

Include photograph of the experimental setup.

We can include some photos but in our opinion is not very relevant for the reader. Two images are presented below: the first one is presenting the overall set-up, while the 2nd picture the cell and the microelectrode employed is depicted. Should you consider these photo relevant, we can include them in the manuscript.

Comparison with other recent works should be included to make clear the contribution of the work. The results must be not only qualitative, but FoMs/metrics to parametrize the achieved detection improvement should be introduced.

The results we have at the time being are from our viewpoint insufficient for performing an accurate quantitative analysis. On the other hand, the aim of the study and the novelty is the one of demonstrating the operability/functionality of DEP+EIS at the level of primary cells (and their differentiation) fact which in our opinion is successfully achieved. This approach, to our knowledge, was not previously reported within scientific literature. However, the results obtained by us are in line with the results reported in the literature and collected and presented by us within the paper “Turcan, I.; Olariu, M.A. Dielectrophoretic Manipulation of Cancer Cells and Their Electrical Characterization. ACS Comb. Sci. 2020, 22, 554–578, doi:10.1021/acscombsci.0c00109.” Besides, Fig. 4 in the manuscript is in accordance with the data previously reported in the literature and in line with the data collected and plotted within Fig. 13 from the paper “Turcan, I.; Olariu, M.A. Dielectrophoretic Manipulation of Cancer Cells and Their Electrical Characterization. ACS Comb. Sci. 2020, 22, 554–578, doi:10.1021/acscombsci.0c00109.”

We hope with all the above responses will be suitable for publication in MDPI Biosensors.

Sincerely yours,

Dr. Marius Andrei Olariu

Round 2

Reviewer 2 Report

Thank you for your responses.

Yes, I think the paper should include all the information for the clarity of the readers. In this sense it is important to justify the differents choices made at the different stages: electrodes, signals (it would be better to unify and work with the same 9Vpp excitation signal), ... I think all these is relevant, since your analysis is qualitative, and you are not going to deepen in the analysis.

Author Response

Dear Reviewer,

As requested by you, within section 2.1 we included info regarding the geometry of the IDEs. On the other hand, we include the info justifying why we chosen 12Vpp voltage within section 3.1. The info modified is highlighted with red font in Manuscript - revision 2.

Moreover, we are sending a Supplimentarry info annex within which the impedance spectra are depicted.

Thank you again for your availability in reviewing our manuscript!

Looking forward to hearing from you!

Warmest regards,
Marius
